# Explore Hybrid Modeling for Moving Infrared Small Target Detection

## ABSTRACT

Moving infrared small target detection, crucial in contexts like traffic management and maritime rescue, encounters challenges from factors such as complex backgrounds, target occlusion, camera shake, and motion blur. Existing algorithms fall short in comprehensively addressing these issues by exploring hybrid modeling, impeding generalization in complex and dynamic motion scenes. In this paper, we propose a hybrid modeling method for moving infrared small target detection via smoothed-particle hydrodynamics (SPH) and Markov decision processes (MDP). SPH can simulate the motion trajectories of targets and background scenes, while MDP can optimize detection system strategies for optimal action selection based on contexts and target states. Specifically, we develop an SPH-inspired image-level enhancement algorithm which models the image sequence of infrared video as a 3D spatiotemporal graph in SPH. In addition, we design an MDP-guided temporal feature perception module. This module selects reference frames, aggregates features from both reference frames and the current frame. The previous and current frames are modeled as an MDP tailored for multi-frame infrared small target detection tasks, aiding in detecting the current frame. Conducted extensive experiments on two public dataset: DAUB and DATR, the proposed network surpasses the state-of-the-art methods in terms of objective metrics and visual quality.

## CCS CONCEPTS

• **Computing methodologies** → **Object detection**; **Matching**.

## KEYWORDS

Moving infrared small target detection, Deep Learning, Mathematical model, Smoothed-particle hydrodynamics, Markov decision processes

## 1 INTRODUCTION

Identifying moving targets in challenging weather conditions such as fog and heavy rain is often difficult with visible light videos. In contrast, infrared (IR) videos offer more reliable target detection, even in adverse weather, due to their unique imaging mechanism [26, 36, 39]. Accordingly, detecting small moving targets, derived from this unique modality, *i.e.*, IR videos, is a prominent

**Unpublished working draft. Not for distribution.**

Permission to make digital or hard copies of all or part of this work for personal or classroom use is granted without fee provided that copies are not made or distributed for profit or commercial advantage and that copies bear this notice and the full citation on the first page. Copyrights for components of this work owned by others than the author(s) must be honored. Abstracting with credit is permitted. To copy otherwise, or republish, to post on servers or to redistribute to lists, requires prior specific permission and/or a fee. Request permissions from permissions@acm.org.

*ACM MM, 2024, Melbourne, Australia*

© 2024 Copyright held by the owner/author(s). Publication rights licensed to ACM.

ACM ISBN 978-x-xxxx-xxxx-x/YY/MM

https://doi.org/10.1145/nnnnnnn.nnnnnnn

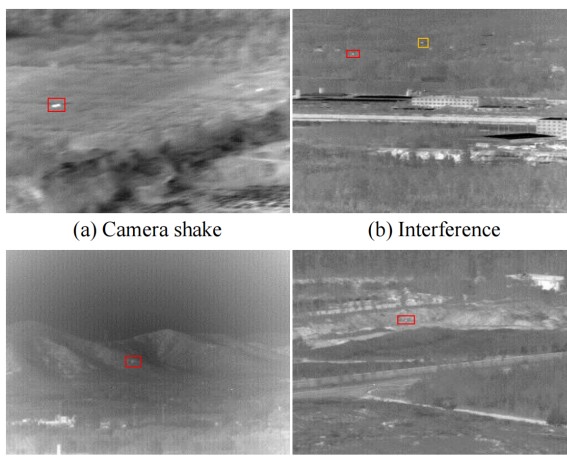

(a) Camera shake      (b) Interference

(c) Motion blur      (d) Complex background

**Figure 1: Typical issue with moving infrared small targets. Small targets marked by red bounding boxes, interferences by yellow.**

subject in computer vision, widely applied in traffic management and maritime rescue [28, 37].

As shown in Figure 1, the long-distance nature of infrared imaging results in insufficient information regarding target details such as size, shape, and texture. Challenges arise for moving infrared targets, including complex background, motion blur, interference and camera shake [9, 21]. Addressing these internal and external factors makes the task of detecting moving infrared small targets in video sequences exceptionally challenging.

In recent years, numerous algorithms have emerged for infrared small target detection [6, 12, 30, 38], categorized into single-frame and multi-frame methods. Single-frame methods focus on small target characteristics, utilizing complex nested network structures and attention modules to minimize information loss during pooling and downsampling processes. DNA-Net [20] employs densely nested interactive and spatial attention modules for feature fusion and enhancement, while UIU-Net [31] achieves multi-level learning by embedding a small U-Net into a larger one, yielding promising results in single-frame detection. However, limitations such as occlusion, motion blur, and camera shake hinder single-frame methods' efficacy in capturing moving infrared small targets. Human visual judgment can infer a blurry target's identity by leveraging information from adjacent frames in videos. Utilizing multiple frames provides rich temporal information compared to a single frame, enabling various multi-frame methods like *image-level target enhancement* and *temporal feature perception* [4, 25, 32, 45].

For image-level target enhancement, Du *et al.* [10] enhance small targets through inter-frame alignment, yet overlook spatial information's significance. Zhu *et al.* [45] leverage optical flow to enhance

moving targets, but this method assumes constant target brightness, which may not hold for infrared small targets due to fluctuations caused by factors like temperature changes and occlusions. In addition, direct application of mainstream YOLO series enhancement algorithms[15, 18, 19] like mosaic enhancement to infrared small targets often leads to target loss. In summary, existing image-level enhancement methods have progressed but have limitations on specific modalities like IR videos, thus failing to achieve desired detection effects. *Can we view image-level target enhancement from a mathematical modeling perspective?* By scientifically modeling the motion state of the target, we can capture more temporal information. Smoothed Particle Hydrodynamics (SPH) simulates fluid behavior by dividing it into particles and simulating their interactions. SPH can model image-level enhancement, simulating information transmission and interaction within IR images, thus facilitating enhancement.

For temporal feature perception, existing methods often employ complex temporal feature aggregation networks. For instance, SSTNet [4] leverages LSTM's memory prediction and a cross-slice ConvLSTM structure to aggregate temporal information. Luo *et al.* [22] utilize dense nested structures and optical flow to design a multi-scale optical flow reconstruction network for capturing moving small targets. The above temporal feature aggregation networks utilize neural networks' nonlinear properties and parameter learning to handle complex temporal data, potentially boosting performance at the cost of increased training expenses. *Can we treat temporal feature perception as a prediction task and model state evolution in time-series IR data?* The Markov decision process (MDP) can offer a simplified approach to modeling time-series IR data by abstracting it into states and corresponding transition probabilities. This can allow us to move away from complex temporal aggregation networks and focus on prediction instead.

In this paper, we propose a method to explore hybrid models for detecting small moving infrared targets using SPH and MDP. Inspired by SPH, we represent motion as fluid dynamics at the image level, with the background and target modeled as stationary and moving particles, respectively. We develop an SPH-inspired image-level enhancement algorithm, using 3D spatiotemporal modeling and SPH Gaussian elliptical kernels for 3D sliding filtering. During sliding, it enhances local contrast and aggregates temporal information, improving efficiency and unifying spatiotemporal dimensions. In addition, we design an MDP-guided temporal feature perception module, comprising a lightweight feature aggregation network and a prediction propagation module. It enriches the temporal information of the target in the current frame by aggregating the reference frame, while reusing detection results from the previous frame and integrating current frame predictions to model the Markov decision. This assists in detecting the current frame across various modeling states and extracting temporal information from multiple frames. Experimental results on two public datasets DAUB[16] and DATR[13], incorporating multiple metrics, indicate that the proposed method outperforms the state of-the-art (SOTA) methods.

·We find hybrid models for moving infrared dim-small target detection, with SPH and MDP playing a crucial role. These models describe target motion and background changes, optimize decision strategies, and boost detection system performance and efficiency.

Experimental results on the DAUB and DATR datasets show that our method surpasses the SOTA methods.

· We pioneer a mathematical approach to image-level target enhancement and design an SPH-inspired image-level enhancement algorithm. Due to SPH's ability to establish strong spatiotemporal relationships, our enhancement algorithm effectively retains details and structure in IR videos, yielding enhanced images with greater accuracy and naturalness.

· We make the first attempt to treat temporal feature perception as a prediction task by designing an MDP-guided temporal feature perception module. This tightly connected and hierarchical module fully exploits temporal information and detection results from reference frames. Modeling the motion process as a Markov model follows explicit and interpretable design principles.

## 2 RELATED WORK

### 2.1 Infrared Small Target Detection in Image

Due to the characteristics of infrared imaging, traditional single-frame detection mainly focuses on modeling the relationship among the target, background, and noise. Traditional algorithms include filter-based methods such as maximum median/mean filters [9] and new top-hat filters [1], human visual system-based local contrast algorithms like LCM [3] and the improved algorithm MPCM [29] based on local contrast, as well as detection algorithms based on sparse representation, such as IPI [14] and improved RIPT [7], etc. However, these methods often lack balance between background suppression and target enhancement and rely heavily on manually extracted features, resulting in lower accuracy and higher false alarm rates.

Influenced by CNN, single-frame infrared small target detection based on deep learning has become mainstream. Dai *et al.* [8] utilize bottom-up attention modulation, integrating low-level features into deeper high-level features. Zhang *et al.* [40] design Taylor finite difference-inspired edge blocks and direction attention aggregation blocks, effectively addressing challenges in detecting the shape of infrared small targets. In addition, Zhang *et al.* [38] try to introduce pruning into small target detection, and used wavelet pruning rules and regularization methods to achieve infrared efficient pruning. Zhu *et al.* [44] design a group of cross stage partial networks and a spatial attention module with global average contrast to obtain local and global spatial semantics. Jia *et al.* [17] abandon the global transformer and the convolutional sliding window of CNN, regarde the local area of the image as a graph node, and apply the graph neural network to infrared small target detection. In order to improve the multi-scale perception ability of the network, Fang *et al.*[12] designed a scale-adaptive feature enhancement mechanism and an attention-guided cross-weighted feature aggregator. While these single-frame detections excel in feature extraction for stationary targets, applying them to moving small targets faces performance limitations due to unique challenges.

### 2.2 Infrared Small Target Detection in Video

Multi-frame detection, with its unique and rich temporal information, outperforms single-frame detection in both accuracy and speed. Traditional multi-frame detection algorithms typically employ methods like energy accumulation. For instance, Zhang *et al.*

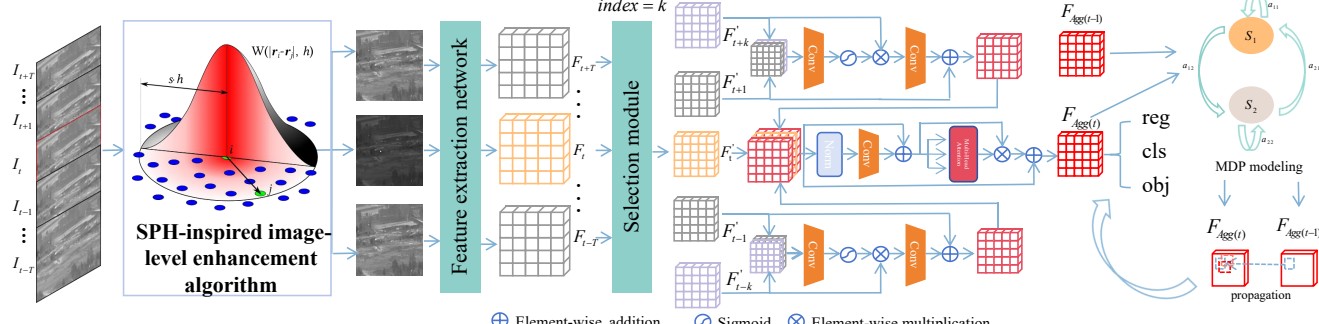

**Figure 2: Overview of the proposed method. On the left is the designed SPH-inspired image-level enhancement algorithm (Section 3.2), which is used in the image preprocessing stage. On the right is the designed MDP-guided temporal feature perception module (Section 3.3), which consists of three parts, inculding the frame selection module, the temporal feature aggregation network and the frame propagation module based on Markov decision modeling.**

[35] simplify the three-dimensional spatiotemporal information into a two-dimensional search, accumulating target energy based on motion direction. Background modeling methods include Zhou *et al.* [41] modeling the current frame and background in the Fourier domain, and Wang *et al.* [27] establishing a spatiotemporal tensor model, representing target extraction as a low-rank and sparse tensor decomposition problem. However, traditional algorithms still face challenges such as poor robustness and high complexity, despite performing well in specific scenarios.

With the development of deep learning, especially in video object detection, the focus of multi-frame infrared small target detection has shifted towards deep learning. The emergence of two-stage Faster-RCNN algorithms [23], and the widespread application of one-stage, YOLO series object detection algorithms have driven advancements in video object detection. FGFA [46] distorts and aggregates adjacent frames onto the current frame through optical flow networks, enhancing target information. MEGA [5] utilizes both local and global temporal information to enhance detection in the current frame. YOLOV [24], based on a one-stage detector, achieves significant success in inference speed by borrowing ideas from region proposals in two-stage detection. TransVOD++[43] proposes a temporal Transformer to aggregate spatial object queries and feature memories of frames. However, these general detection methods excel in learning capabilities for textured medium or large-sized targets but may not universally apply to multi-frame infrared small target detection due to infrared imaging characteristics.

Zhou *et al.* [42] propose an infrared image preprocessing and enhancement algorithm, using techniques like clahe, histogram stretching, and automatic gamma adjustment to enhance each channel separately and extract abundant feature information. But spatial domain enhancement alone is ineffective for moving target scenarios. Yan *et al.* [33] design a multiscale spatiotemporal difference attention network to aggregate more temporal information in feature extraction, achieving a good balance between target discovery and background suppression. Similarly, Bai *et al.* [2] introduce a cross-connected bidirectional pyramid structure and variable ROI pooling to enhance spatiotemporal information. Nevertheless, aggregating temporal information from complex network structures

increases training and prediction costs. To mitigate this, Fan *et al.* [11] combine a lightweight target detection network with target tracking strategies to introduce motion target detection into tracking. Yuan *et al.* [34] design a dedicated module for infrared small target detection and use prior predictions during inference to guide the final output. However, the proposed method only improves upon CIOU but does not fully utilize temporal information during prediction. In summary, the above mentioned method lacks robust mathematical modeling due to the complexity of scenes and variations in this field, hindering comprehensive description with simple mathematical models. This limitation constrains the development of such detection methods, potentially resulting in subpar performance of existing algorithms in real-world applications.

## 3 METHODOLOGY

### 3.1 Overall Architecture

The proposed method, illustrated in Figure 2, is based on the YOLOX [15] framework. Given a set of input frames $Q$ with a range of $2T + 1$, where $Q = \left\{ I_{t-T}, I_{t-(T-1)}, \cdots, I_t, I_{t+1}, \cdots, I_{t+T} \right\}$, before entering the network, it is first modeled as a 3D spatiotemporal graph using the proposed SPH-inspired image-level enhancement algorithm (Section 3.2) for enhancing targets and suppressing backgrounds. At this point, the current frame $I_t$ in $Q$ is significantly enhanced. Subsequently, the entire set of frames is transmitted to a feature extraction network for feature extraction. The feature extraction network adopts an FPN+PAN structure, with all input frames sharing convolutional weights. Later, the output feature set $\{F_i\}, i = t - T, t - T + 1, \cdots t, t + 1, \cdots t + T$, enters the proposed MDP-guided temporal feature perception module (Section 3.3). This module comprises three parts: selection, aggregation, and propagation. Firstly, a selection module picks out a reference frame feature map $F_s$ ($s \in Q, s \neq t$) more effective for the current frame feature map $F_t$. Then, $F_t$ and $F_s$ are jointly input into an aggregation network for fusion, based on a multi-head attention mechanism and multi-scale fusion network. Finally, the propagation module transfers detection results from the previous frame $F_{t-1}$ ($t \geq 1$) to the current frame $F_t$, aiding in current frame detection.

## 3.2 SPH-inspired image-level enhancement algorithm

The local contrast method is commonly used for enhancing infrared small targets, but it typically employs square-shaped kernels and operates on single-frame images. In this study, we introduce an SPH-inspired image-level enhancement algorithm that models sequences as 3D spatiotemporal grids. It replaces square-shaped kernels with Gaussian elliptical kernels from SPH to enhance targets in both temporal and spatial dimensions. Inspired by SPH density fields, this approach is combined with Gaussian difference, as illustrated in Figure 3.

Because of the fixed filter size and variable target sizes, using a square kernel may blend target and background information. Employing a Gaussian elliptical kernel in SPH for local contrast accommodates diverse target sizes. The Gaussian ellipse expression is as follows:

$$\Omega : \frac{(x\cos\theta - y\sin\theta)^2}{(2\sqrt{2}L_{max})^2} + \frac{(x\cos\theta + y\sin\theta)^2}{(\sqrt{2}L_{max})^2} = 1, \qquad (1)$$

where $L_{max}$ denotes the maximum value among all target sizes. $\theta$ represents the rotation angle of the ellipse, with this paper is $\pi/4$.

Inspired by SPH, we regard particles in the fluid as targets and background in the IR video, where the mass of particles corresponds to pixel values. The continuous density field computed by SPH is the ratio of the total mass of particles within a local sampling volume to the volume of the sampling volume. Similarly, we can approximate the pixel value of the central pixel in the elliptical kernel by the ratio of the sum of pixel values within the elliptical kernel to the area of the kernel. The calculation formula is as:

$$I_{avg} = \frac{1}{S_\Omega} \sum_{i=1}^{N} \omega_i I_i, \qquad (2)$$

where $N$ represents all the pixels within the elliptical kernel, $I_i$ is their corresponding pixel values, $S_\Omega$ denotes the area of the elliptical kernel, and $\omega$ stands for the weighting coefficient, which depends on the distance between the pixel and the central pixel.

Subsequently, we divide the elliptical kernel into 9 sub-windows along its major and minor axes. During the sliding process in the spatial dimension, calculate the maximum pixel value $I_{max}$ in the central sub-window. Compute the average grayscale value $g_i$ and the maximum pixel value $g_{max}$ for various sub-windows around the ellipse. The final enhancement for single-image is expressed as:

$$E_t = \min_i \frac{I_{avg}^2}{g_i} \times \varepsilon (I_{max} - G_{max}), \qquad (3)$$

where $\varepsilon$ represents the unit step function. As indicated by the formula, when the target is located in the central sub-window, the target is enhanced at that point.

Based on the rotational symmetry of the Gaussian ellipse, Gaussian ellipse filtering is performed in the temporal dimension. This approach efficiently achieves edge detection and key point detection, aligning features across frames, and ultimately enhancing the target. The final aggregation enhancement formula is as follows:

$$E_{final} = N \left( \sum_{i=t-T}^{t+T} sub(f_{warp}(G_t * E_t - G_i * E_i), E_t) \right), \qquad (4)$$

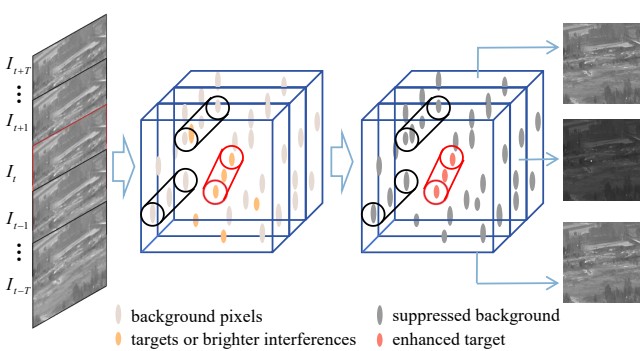

background pixels    targets or brighter interferences
suppressed background    enhanced target

**Figure 3: Overview of the SPH-inspired image-level enhancement algorithm. The black cylinder represents the motion trajectory of the background. It can be observed that, within a short-term frame, the background undergoes no significant changes, while the red cylinder exhibits a noticeable twist, allowing the capture of moving targets using temporal information.**

where $N$ denotes normalization, $sub$ represents background subtraction, $f_{warp}$ signifies feature alignment with the SIFT algorithm, $i$ indicates the reference frame value, and $G$ stands for the Gaussian difference function. The expression is as follows:

$$G(x,y) = \frac{1}{\sqrt{2\pi\sigma_1^2}} exp\left(-\frac{x^2 + y^2}{2\sigma^2}\right). \qquad (5)$$

## 3.3 MDP-guided temporal feature perception module

MDP display memorylessness, where the probability of future states depends only on the current state and is unaffected by past states. This property makes them well-suited for handling sequential data like image sequences, effectively capturing relationships between time intervals. Moreover, MDP effectively model the relationship between past data and future decisions, making them suitable for predicting target positions, states, and other information in time sequences. Accordingly, we can treat temporal feature perception as a prediction task by MPD. And we develop an MDP-guided temporal feature perception module to simplify modeling temporal data by abstracting it into states and corresponding transition probabilities. This reduction in complexity cuts computational costs and enhances interpretation of the model's behavior.

In fact, the MDP-guided temporal feature perception module is divided into three sub-modules: frame selection module, feature-level spatiotemporal information aggregation module, and frame-level prediction information propagation module.

**Frame selection module**. According to the backbone structure of YOLOX, the frame set $Q \in \mathbb{R}^{3 \times T \times W \times H}$, after passing through the feature extraction network, produces outputs for three scales: $F_i^j \in \mathbb{R}^{C_j \times (2T+1) \times W_j \times H_j}$, $j = 1, 2, 3$, Here, $C_j$ comprises three channels with values $[128, 256, 512]$, and $W_j = H_j = [64, 32, 16]$. For efficient reference frame selection, the second scale ($F_i^3 \in$

$\mathbb{R}^{256\times(2T+1)\times32\times32}$) is chosen to match the image scale required by the similarity calculation algorithm.

First, we compute the similarity between frames $F_{t-1}$, $F_t$, and $F_{t+1}$ using the perceptual hash algorithm ($pHash$) as:

$$Sim_{(i)} = pHash\left(F_{t-1}, F_t, F_{t+1}\right), \quad (6)$$

where $pHash$ stands for the perceptual hash algorithm. This algorithm takes fixed-size 32×32 inputs and employs Discrete Cosine Transform (DCT) for pairwise image comparison, yielding the similarity value through Hamming distance.

After obtaining three sub-similarity values, they are normalized to the range (1,$T$), while acquiring the normalization weight $\omega_{nor}$. The expression for $\omega_{nor}$ is:

$$\omega_{nor} = (T-1)/\left(Sim_{(imax)} - Sim_{(imin)}\right), \quad (7)$$

where $\omega_{nor}$ represents the normalized coefficient weight. The final index value is calculated as:

$$index = [1 + \omega_{nor} \times \frac{1}{N}\sum_i Sim_{(i)}]. \quad (8)$$

where [] represents rounding up, $N$ is the total number of obtained similarity values. Frame selection effectively avoids redundancy in spatiotemporal information. High similarity between the current frame and adjacent frames may indicate occlusion, stationary targets, or slow motion. In such cases, the index obtains a larger value, directing attention to more distant and relevant frames. This aligns with human visual perception, extracting richer information from distant frames and eliminating redundant information from similar adjacent frames, facilitating target acquisition. These principles serve as the starting point for designing frame selection modules.

**Feature aggregation module**. After selecting reference frames, the chosen feature set $\left\{F'_i\right\}$, where $i = t-k, t-1, t, t+1, t+k$ ($k = index$), is fed into a lightweight aggregation network for spatiotemporal fusion. This network aggregates features from adjacent frames $F'_{t+1}$ and further reference frames $F'_{t+k}$, followed by fine aggregation with the current frame $F'_t$. The aggregation process resembles a transformer's encoder-decoder structure, incorporating residual connections and multi-head attention modules to enhance temporal features. Due to network symmetry, the explanation is based on one side's structure. Initially, the spatiotemporal information of the distant reference frame $F_{t+k}$ is aggregated with $F_{t+1}$. The aggregation formula is as follows:

$$F^1_{Agg} = \sigma\left[f(F_{t+k}), f(F_{t+1})\right] \otimes f(F_{t+k}) \oplus f(F_{t+1}), \quad (9)$$

where $\sigma$ represents the sigmoid activation function, [] denotes concatenation, $\otimes$ is element-wise multiplication, $\oplus$ indicates element-wise addition, and $f$ stands for the convolution operation. Similarly, $F^2_{Agg}$ is derived from $F_{t-k}$ and $F_{t-1}$. $F^1_{Agg}$ and $F^2_{Agg}$ now represent feature maps obtained by fusing temporal features from adjacent and distant reference frames. Subsequently, they are concatenated with the current frame in preparation for the final temporal feature aggregation. The formula is as follows:

$$F'_{Agg} = \left[F^1_{Agg}, F'_t, F^2_{Agg}\right] \oplus PE, \quad (10)$$

where $PE$ is the added positional encoding, $F'_t$ represents the current frame. The input $F'_{Agg}$ undergoes initial feature extraction through normalization and convolutional layers. The formula is:

$$F''_{Agg} = f\left(Norm\left(F'_{Agg}\right)\right) \oplus F'_{Agg}, \quad (11)$$

where $Norm$ denotes normalization, $f$ represents the convolution operation. Following enhancement processing via the multi-head attention module, we obtain the final feature map $F_{Agg}$, defined as follows:

$$F_{Agg} = \Phi\left(F''_{Agg}\right) \otimes F''_{Agg} \oplus F'_{Agg}. \quad (12)$$

where $\Phi$ represents the multi-head attention module.

**Predictive propagation module**. After feature aggregation, the aggregated frame $F_{Agg}$ is fed into the decoupling head for information prediction. The predicted results, including target coordinates, classification information, and object presence, are then decoded. When $t \geq 1$, we model the motion of small IR targets using an MDP. We utilize the detection results from the previous frame to correct the detection results for the current frame. The MDP consists of quintuplicate variables: $MDP(S, A, P, R, \pi)$. Here, $S$ represents target states, categorized as presence or absence. $A$ stands for the actions performed between states, including detecting the IR target in both frames, not detecting the IR target in both frames, detecting the IR target in the previous frame but not in the subsequent frame, and not detecting the IR target in the previous frame but detecting it in the subsequent frame. $P$ denotes the set of state transitions, while $R$ signifies the reward function, indicating the rewards obtained when different actions are taken in a certain state. The policy $\pi$ defines the actions $A$ that the model may take under various states $S$, along with their corresponding probabilities. By learning each policy, we obtain locally maximal rewards and ultimately achieve the optimal result for the entire detection process. Additionally, we define a dynamic frame variable $k$, storing the most recent IR target detection outcome.

If the IR target is detected in the current frame, we match the policy with the result from the previous frame. The reward function at this point is defined as follows:

$$R^1_{t-1\rightarrow t} = \varepsilon\left(\min_{i,j}\left(\sqrt{\left(x^i_t - x^j_{t-1}\right)^2 + \left(y^i_t - y^j_{t-1}\right)^2}\right)\right), \quad (13)$$

where $i$ and $j$ are the top five highest-scoring results detected in the current frame and the previous frame, respectively. $\varepsilon$ represents the IR target state of the previous frame, defines as follows:

$$\varepsilon = \begin{cases} 1, & if\ box\ exists; \\ 0, & if\ box\ is\ lost; \end{cases} \quad (14)$$

If the IR target remains undetected in the current frame, the reward function is as follows:

$$R^2_{t-1\rightarrow t} = \varepsilon \times \tau_{t-1} + (1-\varepsilon) \times \tau_k, \quad (15)$$

where $\tau_k$ represents the results detected in the most recent frame $k$. Actually, the overall reward function is defined as follows:

$$R_{t-1\rightarrow t} = p[q \times R^1_{t-1\rightarrow t} + (1-q) \times R^2_{t-1\rightarrow t}] + \\ (1-p)[q \times R^2_{t-1\rightarrow t} + (1-q) \times R^1_{t-1\rightarrow t}]. \quad (16)$$

where $p$ and $q$ have the same expression as $\varepsilon$, representing the values of the previous frame and the current frame in two different states $S$.

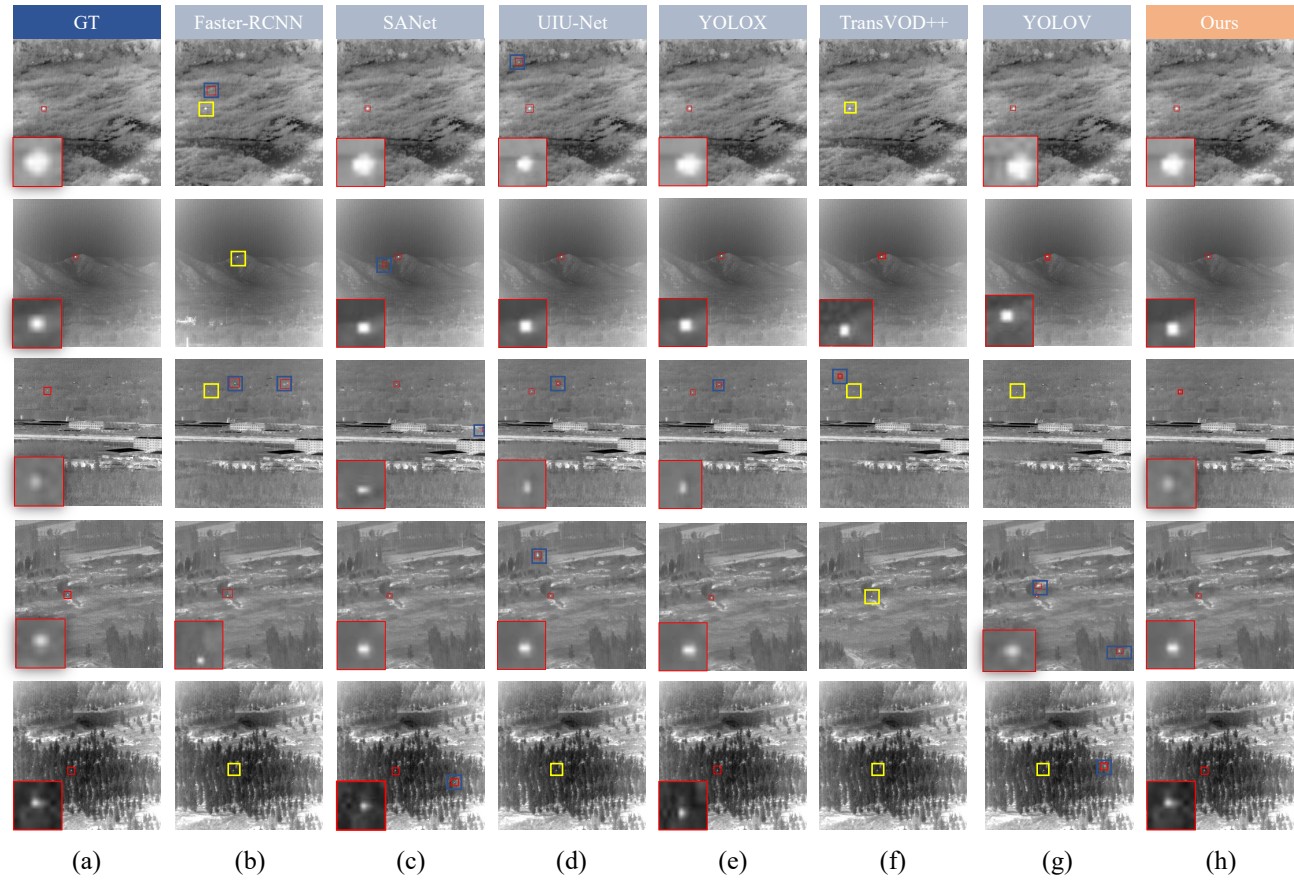

(a)      (b)      (c)      (d)      (e)      (f)      (g)      (h)

**Figure 4: Visual results of different methods. The red boxes represent correctly detected targets, the blue boxes represent false alarms, and the yellow boxes represent missed detections. The first column represents the ground truth and the rest of the columns represent the visual result of Faster-RCNN, SANet, UIU-Net, YOLOX, TransVOD++, YOLOV, and Ours respectively.**

Finally, the adjusted IR target box and score information are compared with the ground truth for loss calculation, and the ultimate loss expression is as follows:

$$L_{total} = \lambda L_{reg} + L_{cls} + L_{obj}. \tag{17}$$

where $L_{reg}$ indicates the regression loss, $L_{cls}$ is the classification loss, and $L_{obj}$ represents the confidence loss.

## 4 EXPERIMENT

### 4.1 Dataset and Implementation Details

*4.1.1 Dataset.* We conduct extensive experiments on the proposed method using two publicly infrared small target datasets DAUB and DATR, along with comprehensive comparative and thorough ablation experiments. The DAUB dataset comprises various scenarios under sky and ground backgrounds, with a total of 22 video sequences. We select 18 sequences that meet the definition of small targets and divide the dataset into a 7:3 training-validation ratio. The training set includes 11 sequences with a total of 9734 frames, while the validation set comprises 6 sequences with 4044 images. The background of the DATR dataset is relatively simple, mainly for tracking and detecting vehicles, but it contains more targets

per frame. The DATR dataset comprises 87 video sequences, with each sequence divided into 250 frames. Sequences 1-76 form the training set with 19000 images, and sequences 77-87 constitute the validation set with 2500 images.

*4.1.2 Implementation Details.* For all experiments, we standardize input images to 512×512 and apply the same data augmentation strategy. During training, we utilize the SGD optimizer with an initial learning rate of 0.01, momentum of 0.937, weight decay of $5 \times 10^{-4}$, and a learning rate reduction factor of 0.1. For DAUB dataset, the maximum training epochs are set to 100, with early termination if performance do not change over multiple epochs, while for DATR dataset, the maximum training epochs are set to 20. The batch size is set to 8, and during the training process, confidence threshold is set to 0.65, and non-maximum suppression is set to 0.3. All experiments are conducted on two NVIDIA RTX-3080 GPUs.

*4.1.3 Evaluation Metrics.* We use object-level evaluation metrics to assess our model's performance, including precision, recall, and F1 score. Precision represents the probability of correct predictions, recall represents the probability of accurate predictions, and the F1 score is the harmonic mean of precision and recall, reflecting the balance between the two. The definition of these metrics are as

**Table 1: Comparison of different methods on the DAUB dataset.**

| Method | Pre(%) | Rec(%) | F1(%) | mAP50(%) |
|---|---|---|---|---|
| UIU-Net[31] | 88.02 | 94.1 | 90.96 | 82.13 |
| DNANet[20] | 93.54 | 96.18 | 94.84 | 89.32 |
| SANet[44] | 92.99 | 96.11 | 94.52 | 83.3 |
| Faster-RCNN[23] | 45.28 | 57.16 | 50.57 | 40.9 |
| YOLOv5[18] | 91.45 | 95.82 | 93.58 | 88.83 |
| YOLOX[15] | 95.93 | 92.95 | 94.42 | 88.97 |
| YOLOv8[19] | 94.2 | 59.4 | 72.86 | 77.26 |
| YOLOv[24] | 91.58 | 80.85 | 85.88 | 72.62 |
| TransVOD++[43] | 83.78 | 65.34 | 73.42 | 54.48 |
| **Ours** | **97.38** | **97.04** | **97.21** | **94.26** |

**Table 2: Comparison of different methods on the DATR dataset.**

| Method | Pre(%) | Rec(%) | F1(%) | mAP50(%) |
|---|---|---|---|---|
| UIU-Net[31] | 98.51 | 93.32 | 96.21 | 92.32 |
| DNANet[20] | 97.13 | 84.39 | 90.31 | 81.36 |
| SANet[44] | 98.39 | 92.55 | 95.38 | 91.63 |
| Faster-RCNN[23] | 75.82 | 88.92 | 81.85 | 66.13 |
| YOLOv5[18] | 89.44 | 95.74 | 92.48 | 84.27 |
| YOLOX[15] | 98.72 | 95.43 | 97.00 | 94.02 |
| YOLOv8[19] | 93.97 | 88.81 | 91.32 | 82.29 |
| YOLOv[24] | 98.56 | 96.12 | 97.63 | 93.60 |
| TransVOD++[43] | 70.65 | 62.91 | 66.56 | 43.50 |
| **Ours** | **99.32** | **97.44** | **97.86** | **96.13** |

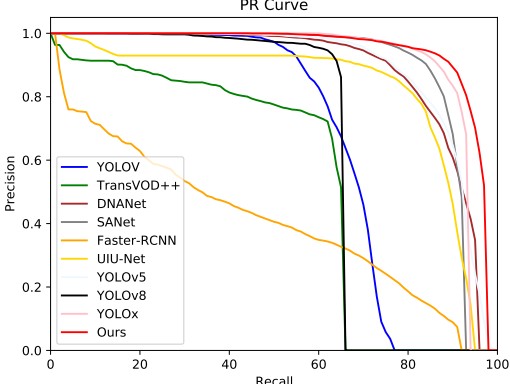

**Figure 5: PR curves of different methods on DAUB dataset.**

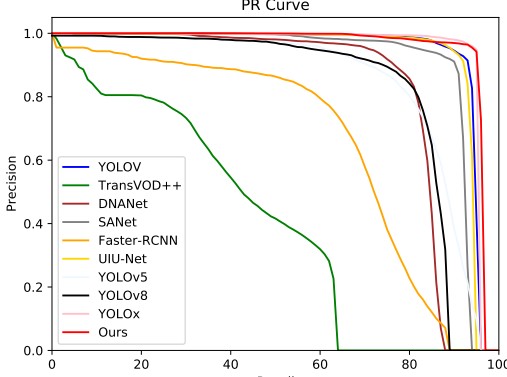

**Figure 6: PR curves of different methods on DATR dataset.**

follows:

$$Precision = \frac{TP}{TP + FP}, \tag{18}$$

$$Recall = \frac{TP}{TP + FN}, \tag{19}$$

$$F1 = \frac{2 \times Precision \times Recall}{Precision + Recall}, \tag{20}$$

where $TP$, $FP$, $FN$ denote the true positive, false positive, false negative, respectively.

In addition, we compute the PR curve of the model. The PR curve reflects the relationship between precision and recall at different confidence levels. From this curve, we derive mAP using the following formula:

$$mAP = \frac{1}{n} \sum_{1}^{n} \int_{0}^{1} precision(recall)d(recall). \tag{21}$$

In this study, $n$ denotes the number of target categories, which is 1. Additionally, we use an IOU threshold of 0.5 for computing mAP, referred to as mAP50.

## 4.2 Quantitative Results

We quantitatively compare the proposed method's performance based on mAP50, precision, recall rate, and F1 score, along with

Precision-Recall (P-R) curves. Tables 1 and 2 present the metrics for the 9 object detection methods, including UIU-Net, DNA-Net, SANet, Faster-RCNN, YOLOv5, YOLOX, YOLOv8, TransVOD++ and YOLOV. Our method consistently outperforms others across all metrics, showing superior detection performance. For instance, on the DAUB dataset, our method improves precision, recall, F1, and mAP50 by 3.84%, 0.86%, 2.37%, and 4.94% over the second-ranked methods(DNA-Net), respectively. Notably, Faster-RCNN performs the poorest, potentially due to excessive candidate box generation. While YOLO series detectors show promising results, image enhancement algorithms in preprocessing may overshadow objects, affecting overall performance. Infrared small target detection algorithms achieve excellent results but are hampered by complex networks, leading to slow training and inference speeds. Video detection algorithms like YOLOV and TransVOD++ exhibit poor performance due to unsuitable temporal feature aggregation networks for infrared small targets. While on DATR dataset, our method likewise achieves the best results on four metrics. It improves precision, recall, F1, and mAP50 by 0.6%, 2.01%, 0.86%, and 2.11% over the second-ranked methods(YOLOX). Compared with the DAUB dataset, most of the methods have achieved better improvement on the DATR dataset, and we believe that the reason is that the small targets in the DATR dataset are relatively larger,

the background is relatively simple. It also has to do with how the dataset is divided.

The Precision-Recall (P-R) curve, illustrated in Figure 5 and 6, is a crucial comprehensive metric. It computes precision and recall at various thresholds, revealing the correlation between them and evaluating the relevance of detection results. A curve closer to the upper right corner signifies superior network performance. In Figure 5, our curve, highlighted in red, notably covers almost all the compared methods in the upper right corner, demonstrating our network's superior balance between accuracy and recall, resulting in the best overall performance. In addition, it can be seen that the curve of Faster-RCNN is the most flat, and the curve of YOLOV and YOLOv8 ends quickly with the increase of recall. The curves of the rest of the methods are not much different, but they are all below our curve as a whole. This is consistent with the above analysis and experimental results.

## 4.3 Visual Results

For a more intuitive comparison of contrastive effects, we select six existing methods for visual comparison with our network. And we choosed several typical scenarios in the DAUB dataset, including mountains, clouds, cities, and forests. As depicted in Figure 4, our method accurately locates small targets without producing missed detections or false positives at the same IOU threshold, and it can be clearly seen that the results of our method are highly consistent with the groundtruth. Faster-RCNN exhibits the highest number of missed detections and false positives. This is in line with the results of quantitative experiments. Among single-frame infrared small target detection networks, SANet achieves the highest accuracy but still encounters false positives and missed detections. In the case of the two video detectors, TransVOD++ demonstrates subpar visual results, possibly due to challenges in training as a transformer detector and limited applicability to multi-frame infrared small target detection as a general detection framework. YOLOV has a good detection effect in simple backgrounds, but missed detections and false detections occur in complex backgrounds and very small target situations. Because of the image preprocessing stage of YOLO causes confusion of complex backgrounds and targets, resulting in submerged targets. YOLOX showed good results, but their detection boxes did not have the best coincidence with the groundtruth.

## 4.4 Ablation Study

To validate the effectiveness of our proposed modules, we conduct ablation experiments, with results shown in Table 3. The base network solely utilizes YOLOX as the detector, without integrating the SPH-inspired image-level enhancement algorithm and MDP-guided temporal feature perception module. We then progressively add these components to YOLOX and observed their impact. Results indicate significant enhancements when incorporating both the MDP-guided temporal feature perception module and the SPH-inspired image-level enhancement algorithm into the base framework. Specifically, on both datasets, the baseline with the SPH-inspired image-level enhancement algorithm improves mAP50 by 1.38% and 1.52%, respectively. Similarly, the baseline with the MDP-guided temporal feature perception module improves mAP50 by 3.07% and 1.94%, respectively. Notably, combining both modules

**Table 3: Ablation experiments for components of the proposed method on DAUB dataset.**

| Method | mAP50(%) | mAP0.50:0.95(%) |
|---|---|---|
| YOLOX | 88.97 | 50.33 |
| +MDP-guided | 92.04 | 52.27 |
| +SPH-inspired | 90.35 | 51.85 |
| +MDP-guided+SPH-inspired | 94.26 | 54.32 |

results in synergistic effects, elevating mAP50 by 5.29% and 3.99%, respectively, demonstrating a greater performance boost than the sum of their individual contributions.

Through ablation experiments, we observe a notable trend: the MDP-guided temporal feature perception module outperforms the SPH-inspired image-level enhancement algorithm. Further analysis reveals that while the SPH-inspired algorithm enhances targets and suppresses background using spatiotemporal information, it faces challenges in detecting small, dark infrared targets against bright backgrounds and clutter. In contrast, the subsequent MDP-guided temporal feature perception module leverages Markov modeling to address moving small targets and eliminate static bright backgrounds, resulting in significantly enhanced detection performance. Essentially, the SPH-inspired algorithm provides coarse localization, reducing false negatives, while the MDP-guided module offers fine localization, reducing both false negatives and false positives. Consequently, comprehensive analysis supports the superiority of the MDP-guided temporal feature perception module in enhancing infrared small targets compared to the SPH-inspired image-level enhancement algorithm.

## 5 CONCLUSION

This paper presents a multi-frame infrared small target detection network by finding hybird models: SPH and MDP. SPH simulates fluid behavior by dividing it into particles and modeling their interactions. It can also simulate information propagation and interaction in images, enhancing them. MDP, known for their memorylessness, is effective for handling sequential data like image sequences, capturing relationships between time intervals. Accordingly, the MDP-guided temporal feature perception module effectively addresses challenges posed by complex backgrounds and occlusions, while the SPH-inspired image-level enhancement algorithm tackles issues arising from camera shake and motion blur. Results on the dataset demonstrate improved accuracy in target detection with reduced false positive and false negative rates. Ablation experiments highlight the contributions of the designed modules and algorithms to enhancing network performance. In the future, we can consider integrating data from different sensors (such as infrared and visible light) to enhance the performance and robustness of target detection. Further algorithm optimization is also essential to minimize power usage and enhance real-time capabilities, meeting the requirements of resource-constrained environments and real-time applications.

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
