# OpenReview forum: "Explore Hybrid Modeling for Moving Infrared Small Target Detection"
_acmmm.org/ACMMM/2024/Conference — MM2024 Poster_

### Official Review · Reviewer_y2jG · 2024-05-13

**Rating:** 2
**Confidence:** 2

**Summary:**

This paper presents a hybrid modeling approach using Smoothed Particle Hydrodynamics (SPH) and Markov Decision Processes (MDP) for mobile infrared small target detection. SPH simulates the motion trajectories of targets and background scenes, while MDP optimizes detection system strategies based on environmental and target states to achieve optimal action selection guided by context and target conditions.

**Strengths:**

- The MDP-guided temporal feature perception module effectively addresses challenges posed by complex backgrounds and occlusions. The image-level enhancement algorithm inspired by SPH tackles issues arising from camera shake and motion blur.

**Limitations:**

This paper has two issues in the experimental section.

1. Although the DAUB dataset and DATR dataset to some extent reflect the model's good performance, there is a shortage of certain datasets, such as NUDT-SIRST and NUAA-SIRST, in the testing phase. Does this lack of data raise concerns about evaluating the model's generalization and robustness?

2. In the selected models, while the model proposed in this paper demonstrates superior performance in results, YOLO, Faster-RCNN, and TransVOD++ are not tailored for tasks such as detecting small infrared targets. For example, DNANet and ISNet, which are designed for the same task, exhibit excellent performance. Does this indicate potential issues with the selection of models for comparison?

**Suitability:**

2

---

### Official Review · Reviewer_ckCv · 2024-05-24

**Rating:** 4
**Confidence:** 2

**Summary:**

The paper proposes a hybrid modeling method for moving infrared small target detection using smoothed-particle hydrodynamics (SPH) and Markov decision processes (MDP). The SPH simulates the motion trajectories of targets and background scenes, while the MDP optimizes detection system strategies for optimal action selection based on contexts and target states. The proposed method includes an SPH-inspired image-level enhancement algorithm and an MDP-guided temporal feature perception module. The experimental results on two public datasets, DAUB and DATR, demonstrate that the proposed method outperforms state-of-the-art methods in terms of objective metrics and visual quality.

**Strengths:**

The paper introduces a novel hybrid modeling approach for moving infrared small target detection, utilizing SPH and MDP, which effectively describes target motion and background changes and optimizes decision strategies.

The proposed SPH-inspired image-level enhancement algorithm effectively retains details and structure in IR videos, resulting in enhanced images with greater accuracy and naturalness.

The paper pioneers a mathematical approach to image-level target enhancement and introduces an MDP-guided temporal feature perception module, which fully exploits temporal information and detection results from reference frames.

**Limitations:**

In Section 3.2, it is mentioned that according to the backbone structure of YOLOX, the frame set 𝑄 ∈ $R^{3×𝑇×𝑊×𝐻}$. It would be beneficial to clarify the significance of "T" in this context and provide details on the number of reference frames input into the module. Additionally, including any ablations conducted would be helpful, as the number of reference frames can reasonably affect the model's performance.

Regarding the SPH-inspired image-level enhancement algorithm, it would be valuable to specify the number of reference frames used in this process and whether the number of references affects the algorithm's performance. If so, providing additional ablations in this area would enhance the understanding of the algorithm's behavior.

In the conclusion part, specifically in line 925-926, the authors mention meeting the requirements of resource-constrained environments and real-time applications. It would be beneficial to provide information on the inference time and resource usage of the proposed method. Additionally, it would be valuable to include comparisons with state-of-the-art methods to demonstrate the effectiveness of the proposed approach in resource-constrained and real-time scenarios.

**Suitability:**

2

---

### Official Review · Reviewer_BmnV · 2024-05-24

**Rating:** 4
**Confidence:** 3

**Summary:**

This paper, propose a hybrid modeling method for moving infrared small target detection via smoothed-particle hydrodynamics (SPH) and Markov decision processes (MDP).

**Strengths:**

1.This paper  develop an SPH-inspired image-level enhancement algorithm which models the image sequence of infrared video as a 3D spatiotemporal graph in SPH.
2.This paper make the first attempt to treat temporal feature perception as a prediction task by designing an MDP-guided temporal feature perception module.

**Limitations:**

1. Lack of comparison in terms of parameter count or inference time.
2. Missing analysis on lambda and T.
3.Apart from the YOLO series, there is a lack of analysis on other types of methods, such as transformer-based methods or those based on key points.

**Suitability:**

2

---

### Official Review · Reviewer_V4tk · 2024-05-26

**Rating:** 2
**Confidence:** 3

**Summary:**

The paper addresses the challenges of detecting small moving infrared targets. This paper proposes a hybrid modeling method for moving
infrared small target detection via smoothed-particle hydrodynamics (SPH) and Markov decision processes (MDP). It achieves good performance on SOTA benchmarks.

**Strengths:**

The paper achieves SOTA performance on existing benchmarks.

**Limitations:**

The paper is hard to follow, the Fig3 is confusing.

The cross-frame interaction is quite simple, i.e., a simple convolutional-based attention mechanism. This is also well-explored in the CV community.

The proposed methods are quite general for generic video applications. The reviewer doesn't see why this is the best choice for infrared images and for small targets. The motivation is not clear

The qualitative is not convincing. The improvement is not visible.

The authors didn't provide the computational cost for the proposed method and for comparisons.

**Suitability:**

2

---

### Meta-Review · Area_Chair_keCT · 2024-06-30

**Recommendation:** Accept (Poster)
**Confidence:** 3

**Metareview:**

This paper received contrasting reviews in origin, WR, BA, BA, WR, which became 2 R, WR, and BA after rebuttal.
The main weaknesses raised by the reviewers regard the low novelty of the method, weak motivations for the design of the proposed approach, and the insufficient/incomplete experimental results, and ablations especially. Other issues regard the computational cost and the request of clarification of some parts of the method and of the parameters used within.

The rebuttal is actually addressing these issues in a reasonable manner, the method does have a theoretical ground representing an original contribution, and new results and estimates are provided as requested by the reviewers.
To the AC and SAC opinions, the rebuttal is satisfactorily addressing the raised remarks.
For these reasons, the paper is considered acceptable for publication at ACM MM 2024 conference.